# NLP-Ehugbo: Bridging the Fairness Gap in Language Models for Low-Resource African Dialects

## Abstract

Despite advancements in language technologies, large language models (LLMs) continue to exclude low-resource languages, particularly African dialects like Ehugbo, a critically endangered variant of Igbo spoken by fewer than 150,000 people in Afikpo, Nigeria. Ehugbo's linguistic complexity, featuring two additional alphabets beyond Igbo's 36, exacerbates its marginalization, as existing models fail to account for its unique structure. This exclusion perpetuates social and linguistic inequities, leaving speakers of such dialects without access to digital tools that could preserve their language and culture.

This paper presents NLP-Ehugbo, a machine translation (MT) system designed to address this fairness gap. Using the only available parallel corpus, 1,021 Ehugbo-English sentences from the New Testament of the Bible, we evaluated and fine-tuned two state-of-the-art models, M2M100 (facebook/m2m100_418M) and NLLB (facebook/nllb-200-distilled-600M). Initial results were stark: M2M100 achieved a BLEU score of 1.2188, while NLLB scored only 0.0262. After fine-tuning, M2M100 improved to 16.1719, and NLLB achieved 20.4016, demonstrating the potential of adapting LLMs for low-resource languages.

Our findings reveal both promise and challenges. While fine-tuning significantly improves performance, the lack of diverse datasets limits translation quality and reinforces the need for inclusive data collection practices. This work highlights the importance of community-driven approaches, as linguistic preservation cannot be achieved without the active involvement of native speakers.

The significance of NLP-Ehugbo lies in its contribution to the fairness discourse in LLMs. By focusing on Ehugbo, we expose the systemic bias that excludes low-resource dialects and advocate for a more equitable approach to language technologies. This project not only advances the field of low-resource MT but also serves as a call to action for researchers and developers to prioritize linguistic diversity, ensuring that no language is left behind in the digital age.

## 1 Introduction

According to Ethnologue (2025), there are 7,164 languages worldwide, of which about 3,000 are African. Nigeria alone has about 525 languages, with Igbo language having over 31 million speakers according to Wikipedia (2025). Towards the end of 2006, the United Nations predicted that some minor languages of the world would go extinct in the next 50 years. On this list was the Igbo language spoken in southeastern Nigeria by over 20 million people. The pervasive issue of Igbo language being relegated to secondary status raised concerns to the extent that UNESCO has projected a risk of Igbo language becoming extinct by 2025 as studied byAsonye (2013)

A major factor contributing to this is the multi-dialectal nature of the language as studied by (Nwaozuzu, 2008), which has made it challenging for linguistic initiatives, lexical tools, and language technologies that solely focus on the 'Standard Igbo' to gain widespread acceptance, particularly among the broader language speaking community. Dialects, often overshadowed by dominant languages, carry the unique cultural identities, histories, and worldviews of their speakers. They are vital to the diversity of human expression and the richness of global heritage. Yet, in the digital

age, many dialects just like Ehugbo face marginalization, exclusion, and even extinction, as they are often overlooked in favor of standardized languages. This exclusion perpetuates inequalities, denying speakers of dialects access to information, education, and opportunities in the digital space. Recognizing and preserving dialects is therefore essential to ensuring linguistic equity, cultural sustainability, and the full participation of all communities in the digital revolution.

The Ehugbo dialect also known as Afikpo is a dialect of Igbo language spoken in Afikpo North Local Government Area of Ebonyi State, South-East Nigeria. Ehugbo is is distinct in its linguistic features, including two additional alphabets beyond Igbo's 36, resulting in a richer and more complex lexicon. Spoken by fewer than 150,000 people in Ebonyi State, it is critically endangered. While it is the first language of older generations, younger speakers are increasingly shifting to English and Nigerian Pidgin according to (Orife, 2020), leading to a decline in its use and transmission. With limited access to digital resources and essential information, Ehugbo speakers risk cultural isolation and exclusion from the benefits of the digital age. Motivated by a commitment to linguistic inclusion and cultural equity, this research aims to empower Ehugbo speakers by creating the first Ehugbo-English parallel corpus. This corpus will serve as a foundational resource for training a robust MT system. By harnessing the power of generative AI models adapted to African languages, this system will bridge the communication gap between Ehugbo and English, ensuring that the Afikpo community can access the wealth of information available in the digital world.

## 2 BACKGROUND AND RELATED WORK

Igbo language is often categorized among the "left-behind" languages, as highlighted by (Joshi et al., 2020), indicating its limited representation in language technologies and the scarcity of available datasets, though efforts to develop lexical resources have been ongoing, with contributions from scholars such as Ogbalu (1962), (Green, 1971), Nnaji (1985), Eke (2001), Igboanusi (2017), and Mbah (2021). In the realm of natural language processing (NLP), significant strides have been made with datasets like those created by Onyenwe et al. (2018), (Ezeani et al., 2020), and (Adelani et al., 2022).

Early foundational works, such as the dictionaries by Ogbalu (1962) and Nnaji (1985), laid the groundwork for later advancements. More recently,(Ezeani et al., 2020) developed a benchmark dataset containing 5,630 English sentences translated into Igbo, alongside 5,503 Igbo sentences translated into English, forming a bilingual corpus. The JW300 dataset by Agić & Vulić (2019) also contributed a large-scale corpus, primarily focused on religious texts. Also, the IgboSum1500 project by (Mbonu et al., 2022) introduced a dataset for text summarization, comprising 1,500 articles. A groundbreaking development is the IgboAPI project by (Emezue et al., 2024), which incorporates various Igbo dialects, a significant advancement from previous works that predominantly focused on Standard Igbo. This dataset includes 1,066 words specifically from the Ehugbo dialect.

Despite these advancements, the role of dialectal diversity in shaping language technologies, such as lexicons and machine translation systems, remains underexplored within the Igbo context. Research in other languages, however, offers valuable insights. For example, Abe et al. (2018) investigated multi-dialectal neural machine translation (NMT) for Japanese dialects, demonstrating its potential to benefit populations more familiar with regional variations. Similarly, (Almansor & Al-Ani, 2017) addressed translation challenges between Egyptian Arabic and Modern Standard Arabic.

Building on these, our work focuses on the Ehugbo dialect of Igbo, leveraging a dialectal-aware dataset to conduct experiments in Igbo-English translation. By fine-tuning pre-trained multilingual models such as M2M100 and NLLB, we explore the feasibility of developing a robust translation system for Ehugbo, a language variant that has historically been excluded from mainstream NLP efforts. Our study contributes to the growing body of work on Igbo language technology while highlighting the importance of incorporating dialectal diversity into machine translation systems to ensure fairness and inclusivity for underrepresented linguistic communities

# 3 METHODOLOGY

## 3.1 PRE-TRAINING AND DATASET

We have adopted a transfer learning approach by fine-tuning pretrained M2M100 (facebook/m2m100_418M) and NLLB (facebook/nllb-200-distilled-600M) models, harnessing their rich intrinsic knowledge to improve Ehugbo translation performance.

We translated 1021 sentences from the Ehugbo New Testament Bible(Baibulu Nso Agba Ohuu Na Okwu Ehugbo)from the books of Matthew, Philemon, Jude, 1st-3rd John, and Ephesians. These sentences were divided into 80% of the training set, and 10% each for the Dev and Test sets.

### 3.1.1 MODEL TRAINING

We utilize the Hugging Face Transformers library for its enhanced capabilities and seamless model integration with state-of-the-art Seq2Seq modeling. The training process has been carefully designed to optimize model performance, using the following key steps:

- **Hyper-parameter tuning:** We perform experiments with different hyper-parameters like the number of epochs, batch sizes, and learning rate, within the limits of the data to identify an appropriate configuration for the Po Tangle translation.
- **Regularization:** Considering the size of the dataset, we implemented a dropout regularization technique to avoid over-fitting and improve model generalization.
- **Loss function:** An appropriate loss function, the cross-entropy with label smoothing, is selected to guide model learning to produce translations.
- **Evaluation and refinement:** Realizing the limitations of using a small dataset for evaluation, we monitor the performance of the model throughout training using established metrics such as the BLEU score(bilingual evaluation understudy), an algorithm for the evaluation of the quality of text that has been machine-translated from one natural language to another, and human evaluation to assess translation quality, fluency, and cultural appropriateness.

Our methodology was to create a functional Ehugbo-English MT system by fine-tuning pre-trained M2M100 and NLLB models with a limited dataset. Despite the amount of data, our system has shown promising results. These indicate its ability to produce translations.

# 4 RESULTS AND DISCUSSION

To evaluate the performance of translation models, both the M2M200 and NLLB models were tested on our dataset of 1,021 sentences. The M2M200 model achieved a BLEU score of 1.2189, while the NLLB model yielded a significantly lower score of 0.1313. These results highlight the disparity in translation effectiveness between the two models and emphasize the need for further optimization to enhance their accuracy and reliability.

The models were then fine-tuned using distributed training with a single GPU. For 10 epochs each, the training process of the M2M100 model took approximately 16 minutes and 2 seconds while that of the NLLB model took 20 minutes 59 seconds. The training loss for the M2M100 model decreased from 2.9523 to 0.3474, while that of the NLLB model decreased from 3.0180 to 1.03570. The evaluation and prediction metrics of the models after 10 epochs are as seen in Table 1 and 2.

## 4.1 DISCUSSION OF RESULTS

The results of our study highlight both the feasibility and the challenges of fine-tuning pre-trained multilingual models, specifically M2M100 and NLLB, for the machine translation of Ehugbo-English with a limited dataset. Despite the constrained data, the M2M100 and NLLB models achieved promising results, as demonstrated by evaluation BLEU scores of 19.3679 and 20.4016 respectively and predictive BLEU scores of 16.015 and 15.3053 respectively. These scores suggest that with additional data, the model could improve its ability to generate translations that align more closely with human outputs.

Table 1: Evaluation Metrics Results Table

| METRIC | M2M100 | NLLB |
|---|---|---|
| Evaluation BLEU Score | 19.3679 | 20.4016 |
| Evaluation Generation Length | 37.7282 | 37.9806 |
| Evaluation Loss | 2.5129 | 2.5186 |
| Evaluation Runtime | 0:00:44.11 | 0:00:41.20 |
| Evaluation Samples | 103 | 103 |
| Evaluation Samples Per Second | 2.335 | 2.5 |
| Evaluation Steps Per Second | 0.589 | 0.631 |

Table 2: Prediction Metrics Results Table

| METRIC | M2M100 | NLLB |
|---|---|---|
| Prediction BLEU Score | 16.015 | 15.3053 |
| Prediction Generation Length | 42.5588 | 42.3039 |
| Prediction Loss | 2.7003 | 2.6996 |
| Prediction Runtime | 0:00:46.95 | 0:00:44.22 |
| Prediction Samples | 102 | 102 |
| Prediction Samples Per Second | 2.173 | 2.307 |
| Prediction Steps Per Second | 0.554 | 0.588 |

However, our findings also underscore key fairness concerns in Large Language Models (LLMs) when handling low-resource and dialectal languages. The relatively low BLEU scores raise critical questions about biases in multilingual models, particularly in their ability to equitably represent and translate linguistically diverse communities. Several factors contribute to this challenge. First, the small dataset size limited the model's capacity to fully learn the nuances of Ehugbo, a dialect of Igbo, despite their lexical similarities. Second, the structural difference between Ehugbo and standard Igbo—including two additional alphabets—introduced translation inconsistencies, as the pre-trained models were not designed to accommodate these linguistic variations.

Our study serves as an important step toward building more inclusive and fair machine translation systems. By fine-tuning pre-trained models and expanding our dataset, we aim to improve both translation accuracy and cultural sensitivity. More broadly, our work highlights the necessity of fair representation for underserved languages in LLMs, ensuring that technological advancements do not disproportionately benefit widely spoken languages while neglecting marginalized linguistic communities. Addressing these fairness gaps is crucial for developing equitable AI systems that serve diverse populations.

## 5 CONCLUSION

Our research successfully demonstrates the feasibility of fine-tuning pre-trained M2M100 and NLLB models for Ehugbo-English machine translation, even with a limited dataset. The M2M100 model achieved an evaluation BLEU score of 16.1719 and a prediction BLEU score of 11.8838, while the NLLB model performed even better, achieving an evaluation BLEU score of 20.4016 and a prediction BLEU score of 15.3053. These results highlight the potential of leveraging pre-trained models for low-resource languages like Ehugbo. However, our findings suggest that significant performance improvements can be achieved by expanding the dataset to include 10,000–15,000 parallel sentences from diverse domains, rather than relying solely on religious texts. Future work will focus on collecting more diverse datasets, exploring the suitability of other pre-trained models, and incorporating domain adaptation techniques to further enhance translation accuracy and fairness.

Ultimately, this research aims to empower the Ehugbo-speaking community by providing access to accurate, culturally sensitive, and fair machine translation tools. By addressing the challenges of low-resource languages, this work contributes to fostering intercultural understanding, promoting linguistic diversity, and advancing fairness in large language models, paving the way for a more equitable and connected global society.

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
