# OpenReview forum: "NLP-EHUGBO: BRIDGING THE FAIRNESS GAP IN LANGUAGE MODELS FOR LOW-RESOURCE AFRICAN DIALECTS"
_ICLR.cc/2025/Workshop/BuildingTrust — Submitted to BuildingTrust_

### Official Review · Reviewer_yGRw · 2025-02-22
**NLP-EHUGBO finetunes multilingual machine translation models with small corpuses of available text written in Ehugbo.**

**Rating:** 2
**Confidence:** 4

**Review:**

**Strengths:**
- **Valuable Topic:** Addresses the gap in machine translation for low-resource languages, specifically Ehugbo.
- **Fairness Focus:** Highlights the social and linguistic inequities stemming from underrepresentation in language technologies.

**Weaknesses:**
- **Limited Dataset:** Relies on a small, domain-specific corpus (1,021 New Testament sentences) that fails to capture Ehugbo’s full linguistic diversity.
- **Narrow Evaluation:** Depends solely on BLEU scores without incorporating human assessments or additional metrics in other machine translation works. They also evaluate on the same dataset they finetune on.
- **Methodological Simplicity:** This work offers minimal innovation beyond standard fine-tuning of existing pre-trained models.
- **Insufficient Analysis:** The work does not thoroughly address how Ehugbo’s unique alphabets affect OOD model performance or the impact of the domain-specific corpus on other evaluation.

---

### Official Review · Reviewer_nQeN · 2025-03-02
**NLP-Ehugbo -- Needs more details about experimental results**

**Rating:** 5
**Confidence:** 3

**Review:**

## Contribution
Introducing a machine translation system for Ehugbo-English via fine-tuning existing LLMs.
## Strengths
- There is an ethical importance to increasing the prevalence of marginalized languages in LLM training
- Fine tuning results show promising improvements from the baseline

## Weaknesses
- Many of the training details could be placed in an appendix, or shown via a graph/plot instead of a paragraph description in the results
- Uses BLEU as only evaluation criteria, although BLEU can be quite brittle and non-representative
- Doesn't include any example translation sentences vs. ground truth (hard to grasp improvement just based on the delta in a single number)

## Minor Edits
- Period missing at the end of page 2

## Questions
- Given the sparsity of Ehugbo-specific data, did you consider using other Igbo-dialect datasets in addition to Ehugbo for finetuning, or using some of the Igbo datasets? It would be interesting to see whether finetuning even with those datasets which, although not the same as Ehugbo, are more similar than the majority of the data that these LLMs are trained on, would improve performance for Ehugbo as well.

## Overall Thoughts
This paper is interesting, and addresses an important dearth in LLM training. However, the paper would benefit greatly from more detailed analysis of results beyond just reporting the BLEU score and the training details.

---

### Official Review · Reviewer_AGk3 · 2025-03-02
**The paper addresses the fairness gap of LLM in low resource languages like Ehugbo. The authors fine-tuned the pre-trained Ehugbo-English models by expanding the dataset.**

**Rating:** 4
**Confidence:** 3

**Review:**

Strengths:
1. The paper address an important issue of culturally sensitive and fair LLMs for low-resource languages
2. The authors have expanded the dataset and fine-tuned the pre-trained models for machine translation.

Weakness:
1. The main contribution is the dataset the authors have not discussed the quality of the dataset and the dataset creation itself.
2. The authors did not compare the fine-tuning on the dataset with any SOTA transfer learning approaches or other methods.
3. The novelty is clearly missing in the paper
4. The authors may experiment on low resource language learning techniques such as back-translation,unsupervised data augmentation etc

---

### Decision · Program_Chairs · 2025-03-04

Reject